# Interaction between Fish Skin Gelatin and Pea Protein at Air-Water Interface after Ultrasound Treatment

**DOI:** 10.3390/foods11050659

**Published:** 2022-02-23

**Authors:** Davide Odelli, Krystalia Sarigiannidou, Alberto Soliani, Rodolphe Marie, Mohammad Amin Mohammadifar, Flemming Jessen, Giorgia Spigno, Mar Vall-llosera, Antonio Fernandes de Carvalho, Michela Verni, Federico Casanova

**Affiliations:** 1Research Group for Food Production Engineering, National Food Institute, Technical University of Denmark, SøltoftsPlads, 2800 Kongens Lyngby, Denmark; davide.odelli@ufv.br (D.O.); krystaliasari@gmail.com (K.S.); albertosoliani97@gmail.com (A.S.); moamo@food.dtu.dk (M.A.M.); fjes@food.dtu.dk (F.J.); 2Department for Sustainable Food Process (DiSTAS), Università Cattolica del Sacro Cuore, Via E. Parmense 84, 29122 Piacenza, Italy; giorgia.spigno@unicatt.it; 3Departamento de Tecnologia de Alimentos, Universidade Federal de Viçosa (UFV), Vicosa 36570-900, Brazil; antoniofernandes@ufv.br; 4Department of Health Technology, Technical University of Denmark, Ørsted Plads, 2800 Kongens Lyngby, Denmark; rcwm@dtu.dk; 5Department of Biology and Biological Engineering, Food and Nutrition Science, Chalmers University of Technology, SE-41296 Gothenburg, Sweden; marvallju@gmail.com; 6Department of Soil, Plant, and Food Sciences, University of Bari “Aldo Moro”, 70126 Bari, Italy; michela.verni@uniba.it

**Keywords:** foaming properties, fish skin gelatin, pea protein, interfacial properties, Turbiscan Tower, CLSM

## Abstract

The interaction between fish skin gelatin (FG) and pea protein isolate (PPI) was investigated at the air-water interface (A-W) before and after a high intensity (275 W, 5 min) ultrasound treatment (US). We analyzed the properties of the single protein suspensions as well as an equal ratio of FG:PPI (MIX), in terms of ζ-potential, particle size, molecular weight, bulk viscosity and interfacial tension. The foaming properties were then evaluated by visual analysis and by Turbiscan Tower. Confocal laser scanning microscopy (CLSM) was employed to explore the role of the proteins on the microstructure of foams. The results showed that the ultrasound treatment slightly influenced physicochemical properties of the proteins, while in general, did not significantly affect their behavior both in bulk and at the air-water interface. In particular, PPI aggregate size was reduced (−48 nm) while their negative charges were increased (−1 mV) after the treatment. However, when the proteins were combined, higher molecular weight of aggregates, higher foam stability values (+14%) and lower interfacial tension (IFT) values (47.2 ± 0.2 mN/m) were obtained, leading us to assume that a weak interaction was developed between them.

## 1. Introduction

Foam can be described as a two-phase system in which gas bubbles are uniformly dispersed into a continuous liquid phase. Gas bubbles are separated by a thin continuous layer of liquid called the lamellar phase [1]. In food products, this system could be very complex, containing different mixtures of gases and liquids, which contribute to the texture and the palatability of foods. In order to prevent their agglomeration and coalescence that can cause their collapse, different type of surfactant could be added to ensure foam stability during the time. Thanks to their hydrophilic and hydrophobic groups, these elements can stabilize the interface interacting with the liquid and the gas phases simultaneously. Between food surfactants, proteins are largely employed in food industry. Due to their ability of being adsorb at the air-water interface, proteins can significantly affect the foaming properties and, thanks to their different structures, vary in their behavior: ideal protein foaming agents should be able to rapidly stabilize the system at low concentrations and over different pH range. Egg white, dairy, gelatins, gluten and soy proteins are the main surfactants used in food foams due to their high efficiency [2]. 

However, more sustainable proteins sources have been investigated [3]. Due to its special properties, such as gelation, film forming ability and interfacial properties, gelatin is widely employed in food, pharmaceutical and cosmetic industries to improve stability, elasticity, and texture of many products [4,5]. Gelatin is a denatured and biodegradable protein derived from either alkaline or acid partial hydrolyzation of collagen [6]. Recently, the biochemical and physico-chemical properties of gelatin from saithe fish skin, has been investigated by Casanova et al. [7] to stabilize foams. The authors investigated the foaming properties of fish skin gelatin after different combinations of time and high intensity ultrasound treatment. 

In the last years, pea protein isolate extracted from *Pisum sativum*, which is the European most cultivated protein source, has been employed in the food industry as a functional ingredient [8,9]. Its interfacial properties are mainly given by vicilin, legumin and convicilin, globular proteins from which it is composed. However, due to their large and compact structure, their ability to adsorb at the air-water interface can be limited [10]. Different methods to improve PPI functional properties were investigated, such as pH and/or heat treatment and polysaccharide addition. Xiong et al. [10] tried to improve their foaming properties by applying a high intensity ultrasound treatment resulting in an encouraging enhancement in foam stability. Hinderink et al. [11] tried to improve the interfacial properties of pea protein isolate by blending it with dairy proteins. The authors concluded that the interfacial properties of individual proteins are not additive, but they can react giving stronger or more elastic layers, which can affect the product stability at the interface. The aim of this study is therefore to investigate the behavior of fish skin gelatin (FG), pea protein isolate (PPI) and their mixture (MIX) in three different ratios (100:0, 50:50 and 0:100) in bulk and at the air-water interface before and after ultrasound treatment. 

## 2. Materials and Methods

Fish skin gelatin (FG) was purchased from Sigma Aldrich (Sigma, St. Louis, MO, USA) while F85F Pea Protein Isolate (PPI) was kindly donated by Roquette Frères (Lestrem, France). The composition of FG and PPI powders (i.e., moisture, ash, protein content, mineral composition) and in solution (i.e., thermal properties and surface hydrophobicity) were already assessed by a previous study conducted by Vall-llosera et al. [12]. A phosphate-buffered saline solution (PBS buffer at pH 7.2–7.4) was used for the preparation of the samples. The solution was prepared in double-distilled water with a concentration of 50 mM Na_2_HPO_4_/NaH_2_PO_4_ and 150 mM of NaCl. 

### 2.1. Samples Preparation 

FG powder was dissolved into 200 mL of PBS buffer at a concentration of 18 g/L and stirred for 24 h at room temperature (20 ± 1 °C) to reach a complete solubilization of the protein. PPI solution was stirred for at least 48 h at 4 °C according to Hinderink et al. [11]. The 50:50 ratio solution (MIX), was prepared mixing 100 mL of both the protein solutions maintaining the same concentration. All the solutions were stored in blue caps glass containers (250 mL volume). Sodium azide (Sigma, St. Louis, MO, USA) at 0.02% was added to prevent microbial activity. 

#### Standardization of the FG and PPI Solutions

Since PPI powder presents 80% of protein, 5 solutions at different concentrations (100, 110, 120, 150 and 200 g/L) of PPI were prepared in PBS buffer and stored at 4 °C under stirring for 48 h. The solution was then centrifuged at 12,298 g for 1 h. The supernatant was then evaluated in terms of soluble protein content by Dumas method, using a nitrogen conversion factor of 6.25. A calibration line (Appendix A) was designed to standardize the supernatant solution at 18 g/L of soluble protein, which was obtained with a starting concentration of 110 g/L. 

### 2.2. Ultrasound Treatment 

An aliquot of 50 mL of FG, PPI and MIX was treated with ultrasounds (Branson, Danbury, CT, USA) into a 100 mL glass beaker. To prevent the increasing temperature due to the treatment, the samples were located in an ice bucket for the whole treatment period. The samples were sonicated in a continuous mode at a 20 kHz frequency, 275 W (50% of amplitude) for 5 min. 

### 2.3. pH and Temperature Values

The pH of all the samples was measured before and after the treatment at room temperature (20 ± 1 °C), by an automated pH meter (Metrohm 780, Smedeland, Glostrup, Denmark). The pH values are reported as the average of three repeated measurements and their standard deviations.

### 2.4. Hydrodynamic Diameter (Dh)

The particle size of the untreated and sonicated solutions was measured using a Zetasizer Nano Series (Malvern Instruments, Malvern, UK) instrument. For the DLS analysis, the solutions were diluted 1:10 with Milli-Q water before the injection in capillary cells and analyzed with a wavelength of 633 nm and a scattering angle of 173°. The particle size values were calculated according to the Stokes-Einstein equation as follows:(1)$$D_{h}=\frac{K_{B}T}{3\pi \eta D_{t}}$$
where *D_h_* is the hydrodynamic diameter of the particles, *D_t_* is the diffusion coefficient, which was extracted from the fit of the correlation curve using the cumulative method, *K_B_* is the Boltzmann’s constant, *T* the temperature and η the solvent viscosity (Pa s^−1^). The *D_h_* is hereby reported as the average of three repeated measurements and their standard deviations. 

### 2.5. ζ-Potential Measurements

ζ-Potential of the particles was determined by applying a voltage of 50 V. The values were calculated with the Henry equations as follows:(2)$$\zeta =\frac{3\eta \mu }{2\varepsilon f\left(\kappa R_{h}\right)}$$
where *μ* is the electrophoretic mobility (V Pa^−1^ s^−1^), *η* the solvent viscosity (Pa s^−1^), *ε* is the medium dielectric constant (dimensionless), *κ* is the Debye length or the thickness of the double electric layer around the molecules (nm) and *R_h_* is the hydrodynamic radius (nm). A value of 1.5 was used for *f*(*κR_h_*), which is the Henry’s constant, according to the Smoluchowski approximation [13] since the measurements were conducted in an aqueous medium. 

### 2.6. SEC-MALS

Molecular weight of PP, FG and MIX, before and after ultrasound treatment was determined using size-exclusion chromatography. For that purpose, 9 mg/mL sample were prepared in phosphate buffer (pH 7.2) and filtered with 0.1 μm pore size filter. The HPLC (Agilent, Santa Clara, CA, USA) was equipped with WTC-015S5 column (300 × 7.8 mm, 150 Å maximum pore size) Wyatt Technology, Santa Barbara, CA, USA). The elute was monitored by a UV detector at 280 nm, a DAWN 8 light-scattering detector (Wyatt Technology, Santa Barbara, CA, USA) and an Optilab differential refractometer (Wyatt Technology). The flow rate was 0.8 mL/min and the injection volume 50 μL. The mobile phase was phosphate buffer, pH 7.2. 200 μL/L proClin (Sigma, St. Louis, MO, USA) was added to prevent the microbial growth. The buffer was prior filtered with a sterile single use vacuum filter (Thermo Fisher Scientific, Roskilde, Denmark) with pore size of 0.1 μm. Data analysis and molecular weight calculations were performed using the ASTRA software (7.3.2 Version, Wyatt Technology Europe, Dernbach, Germany).

### 2.7. In Bulk Rheological Analysis 

The rheological properties of the solutions were evaluated with a controlled-stress rheometer (StressTech HR Cannon instruments, State College, PE, USA) equipped with a double gap geometry. A volume of 15 mL was analyzed 20 °C. Oscillatory measurements were used as a first step to find the linear viscoelastic region (LVR), which indicates a Newtonian behavior. LVR was found within a shear rate between 1–100 s^−1^, which was used for the flow measurements. Flow data were fitted with the Power-law model and the apparent viscosity of the sample solutions was obtained using the following equations:*τ* = *m γ ^n^*(3)
*μ_a_* = *m γ*
^*n*−1^(4)
where *τ* is the shear stress, *m* the consistency coefficient, *γ* the shear rate, *μ_a_* the apparent viscosity, and *n* is the flow behavior index. Fluids can be described by three different behaviors depending on *n* value: shear thinning (*n* < 1), shear thickening (*n* > 1) and Newtonian fluids (*n* = 1). Over the shear rate range applied, all of the sample solutions displayed Newtonian flow behavior. Frequency sweep with a frequency range between 1–10 Hz was implemented to determine viscoelastic behavior of the solutions at a 5% fixed strain value. 

### 2.8. Foaming Properties

A volume of 50 mL of solution were put into a glass beaker of 100 mL and foams were produced by Ultraturrax (Colonial Scientific, DI 25 basic yellow line, Richmond, VA, USA) at 9500 rpm for 1 min. Immediately after the whipping process, the foams were poured into a 100 mL graduated cylinder and sealed with parafilm. Foam capacity (*FC*), foam stability (*FS*) and liquid fraction (*LF*), were evaluated by a visual analysis, according to the following equations: (5)$$FC=\frac{V_{t-}V_{0}}{V_{t}}\times 100\%$$
(6)$$FS=\frac{V_{t\;time\;}}{V_{0}}\times 100\%$$
(7)$$LF=\frac{V_{0}-Vt_{liquid}\;}{V_{0}}\times 100\%$$
where *V_t_* represents the volume of the foam after homogenization, *V*_0_ is the initial volume of the protein solution, *V_t time_* and *V_t liquid_* the volume of the foam and the volume of the drainage liquid, respectively, after 0, 5, 10, 15, 20, 30, 45, 60, 75, and 90 min. All the experiments were conducted in triplicate and conducted by the same person at room temperature (20 °C ± 1).

### 2.9. Bubble Size with Turbiscan Tower 

Bubble size was evaluated with Turbiscan Tower (Formulaction, Toulouse, France). The analysis is based on Static Multiple Light Scattering principle. Briefly, a beam of light at 880 nm is sent on the sample and two different detectors acquired the value of the Backscattering (BS) and Transmission (T) all over the sample height. These values depend on the dispersed particle in a sample that are able to scatter the light source. In the case of foam, the scattering caused by bubbles is evaluated. By monitoring the sample over time, the instrument gives an evaluation of the evolution of bubbles and therefore on foam stability. According to Mie Theory [14], it is possible to calculate the size of the bubbles as follows:(8)$$\mathrm{BS}=f\;(\varphi ,\;d,\;n_{p},\;n_{f})$$
where BS is the value of backscattered light, φ is the particle concentration (air fraction in the case of a foam), d is the diameter of the particles, *n_p_* and *n_f_* are the refractive index of the dispersed and continuous phase, respectively. Immediately after the whipping process, the foams were transferred into a 55 mm high tube and loaded into the instrument. The measurements were conducted for 90 min with a scan on the sample every 1.5 min. 

### 2.10. Confocal Laser Scanning Microscopy (CLSM) 

Confocal laser scanning microscopy (CLSM) was utilized to confirm the presence of a protein layer at the air-water (A-W) interface. The solutions were initially diluted to 9 g/L. Rhodamine B (Sigma Aldrich, Gillingham, UK) was used to dye the proteins before following the same whipping process with Ultraturrax for 1 min at 9500 rpm. Immediately after the foam’s formation, a volume of 0.5 mL of each sample was loaded into an 8-chambers microscope slide and covered with a coverslip. The dye was excited at 556 nm and a movie of each sample was taken using a 100× lens (Nikon CFI) on a confocal microscope spinning disc constituted by an inverted microscope (Nikon Ti_2_). This was equipped with a laser source (405/488/561/640 nm), a confocal spinning disc module (Yokogawa CSU-W1, 50 um pinholes), a quad-band emission filter (440/521/607/700 nm) and a sCMOS camera (Photometrics Prime95B). However, due to the relative instability of FG foams, only PPI and MIX samples were analyzed by this technique. Indeed, FG foams drained too much liquid into the sample well, covering all of the air bubbles and hiding them from the microscope vision.

### 2.11. Interfacial Properties 

The interfacial tension (IFT) between the liquid phase (pure water and protein solutions) and the gas phase (air) was measured with an optical tensiometer OCA 25 (DataPhysics Instruments, Filderstadt, Baden-Württemberg, Germany) using a static pendant drop method. IFT was recorded for 1000 s at room temperature (20 ± 1 °C), using a constant drop volume of 20 μL. The IFT values of the solutions were calculated based on the Laplace equation monitoring the shape of the droplet. 

### 2.12. Statistical Analysis 

The mean differences with ±standard deviation (SD) were analyzed. Statistical analysis was performed using student’s test t and one-way analysis of variance test (ANOVA), with a level of significance *p* < 0.05. 

## 3. Results and Discussion

### 3.1. pH Values

Table 1 enlists the pH values of all the sample and a comparison between untreated and sonicated ones. FG did not present any significant changes in pH values after the US treatment, stabilizing around 7.1, while PPI showed a slight decrease from 7.3 to 7.2. As expected, MIX sample exhibited a pH value range between FG and PPI values, which showed a slight decrease after sonication, going from 7.2 to 7.1. These results are in agreement with O’sullivan et al. [9]. The authors described a significant pH decrease of all the US treated animal and vegetable proteins considered, including bovine gelatin, fish gelatin, egg white, soy, rice and pea protein isolates. This is mainly due to the deprotonation of acidic amino acid residues contained within the agglomerated structure of the untreated proteins [15].

### 3.2. Hydrodynamic Diameter (D_h_) and ζ-Potential 

Figure 1 shows the mean hydrodynamic diameter of the sample before and after US treatment. The large standard deviation associated with PPI are in accordance with the results obtained in previous studies on PPI conducted by O’Sullivan et al., and Xiong et al. [9,10]. The authors found a bimodal size distribution: one population has a similar size as the untreated protein (159.07 ± 66.78 nm) while the other one is significantly reduced by the US treatment. As expected, PPI size was reduced after the treatment (112 ± 47.49 nm). A similar behavior was observed for the MIX sample, going from 160.23 ± 69.5 to 115.67 ± 47.40 nm. This trend could be explained by the significant difference in the size between PPI and FG. The reduction of the size is probably due to the disruption of non-covalent associative forces, such as hydrogen bonds and hydrophobic interactions due to the cavitation effect of ultrasound waves [9]. However, FG exhibits a slight increase in the aggregate size after sonication, going from an average of 16.38 ± 8.64 nm to 23.03 ± 17.41 nm. This increase in the size could be due to a thermal aggregation correlated to non-covalent associative forces [9]. ζ-potential measurements present negative values for all of the samples, as presented in Figure 1. In particular, PPI and MIX have a similar tendency to decrease in value after the US treatment. Furthermore, they show similar values. ζ-potential of PPI range from −7.2 ± 0.5 mV to −8.2 ± 0.4 mV, the MIX from −6.4 ± 0.9 mV to −8.0 ± 1.2 mV while FG does not present a significant change in the values, which are stable between −1.32 ± 0.6 mV and −1.28 ± 0.5 mV. 

### 3.3. SEC-MALS

Size exclusion chromatography (SEC-MALS) was employed to determine the molecular weight (Mw) of FG, PPI and MIX prior and after sonication. The obtained results are shown on Table 2. Untreated FG exhibited fraction with Mw lower than expected, as the typical α-chain (~120 kDa) and β-chain (~200 kDa) were not observed. This might be associated to the extensive hydrolysis of collagen throughout the production process of protein [12]. Untreated PPI showed a high Mw fraction of 820 kDa, which is probably corresponded to protein aggregates, while fraction of 164.4 kDa is attributed to the trimeric form of vicilin [16]. Lower Mw fractions (61.2 and 44.1 kDa) are corresponded to vicilin and legumin subunits [16]. Vicilin monomer with Mw of 50 kDa and legumin monomer of 60 kDa had been reported in previous studies [16,17]. Untreated MIX demonstrated higher Mw fractions compared to FG and PPI (Table 2). Ultrasound treatment had no effect on Mw of FG and on lower Mw fractions of PPI. However, it was observed a decrease on Mw of high Mw aggregation of PPI. O’Sullivan et al. [9] reported no difference on molecular structure of FG and PPI after sonication (20 kHz, 95% of amplitude, for 2 min). Sonicated MIX showed higher Mw fraction compared to untreated MIX. The formation of aggregates in protein mixture caused by sonication is in accordance with the study reported by Silva et al. [18], who studied the effect of low-frequency ultrasound (20 Hz, 40% of amplitude, for 1–10 min) on whey-casein mixture.

### 3.4. Rheology 

Figure 2 shows the flow behavior of all the samples, untreated and sonicated, at 20 °C. The curves are presented as shear stress (τ) versus shear rate (γ). Data were evaluated with the power-law model and showed a nearly Newtonian flow behavior (n = 1) for all the samples over the shear rate range applied. Therefore, the consistency coefficient, which is equal to the apparent viscosity in Newtonian fluids, was compared for all the solutions and their values can be found in Table 3. Values were similar for all the samples, varying between 1.0 and 1.5 mPa.s. In particular, sonicated FG and PPI presented a viscosity of 1.0 mPa.s, decreasing compared to their untreated controls which showed the highest viscous behavior of 1.3 and 1.5 mPa.s, respectively. On the other hand, this decrease of viscosity with the US treatment did not occur for the MIX samples, which instead slightly increased from 1.2 to 1.3 mPa.s with the sonication. Therefore, MIX showed a viscosity between FG and PPI suggesting both protein sources contributed to its flow properties. Overall, data are in agreement with O’Sullivan et al. [9] who found a significant decrease of FG (from 1.06 ± 0.07 to 0.76 ± 0.05 dL/g) and PPI (from 0.8 ± 0.005 to 0.76 ± 0.007 dL/g) viscosity after an US treatment. In addition, MIX behavior could be explained by the complexity of the solution which is composed of a mixture of protein fractions rather than single components as FG and PPI [9]. 

A frequency sweep test was also employed to investigate the viscoelastic properties of the protein solutions. However, high concentrated protein solution, as the ones used for this test (18 g/L), protein molecules tend to adsorb at the A-W interface forming a viscoelastic film. Due to this effect, the measured apparent viscosity is the sum of both bulk and interfacial signals [19]. The interfacial protein layer contribution is well seen from the frequency sweep test (Appendix A). The test was conducted into a frequency range between 1–10 Hz and the viscoelastic moduli G′ and G″ were evaluated, where G′ represents the elastic behavior while G″ the viscous behavior of the fluid [20]. At low frequencies, viscous properties are predominant (G″ > G′) while at the end of the considered frequency range, the elastic modulus is definitely higher than the viscous one. 

Protein aggregation and adsorption at the A-W interface contribute to the viscoelastic properties of the fluid giving a gel-like microstructure [21]. Finally, the crossover frequency of the moduli was analyzed. While both of FG samples exhibit frequencies over 4 Hz, PPI samples present a difference between the untreated and the sonicated groups. Untreated PPI shows a crossover around 4 Hz while after the treatment it happens around 1.5 Hz. This could be due to improved protein-protein interactions with the sonication, resulting in a previous manifestation of elastic behavior [21]. On the contrary, MIX samples exhibit the higher frequency value crossover, over 6 Hz for both untreated and sonicated, suggesting a complexity in the aggregates and interfacial film formation. 

### 3.5. Foaming Properties

Foaming properties of protein solutions are generally based on the influence of many factors such as the protein nature, their isolation process, temperature, pH, concentrations, whipping method and time [22]. 

#### 3.5.1. Foam Capacity (*FC*)

*FC* is defined as the percentage fraction of air into the total foam volume [1]. *FC* of all the samples was determined immediately after the whipping process and is here presented as the mean of three replicates. *FC* values for FG, PPI and MIX are presented in Figure 3. We can observe that *FC* improved for sonicated FG in comparison with the untreated sample. The treatment increased the value from 46.6 ± 1.7% to 52.1 ± 1.4%, allowing FG foam to incorporate more volume of air. On the other hand, US treatment did not result in a significant difference for sonicated PPI foams compared to the control, while it slightly reduced *FC* values (−0.86%) of MIX samples. These results disagree with the study carried out by Morales et al. [23]. Indeed, the authors found a significant increase (+62%) of *FC* values for soy protein isolate foams after 5 min of a high intensity ultrasound treatment. However, these results could be explained by the different nature of soy and pea proteins. Overall, the samples reached *FC* value around 50% except for untreated FG which occurred with the lowest value (46.58 ± 1.7%) in all of the three replicates implemented. 

#### 3.5.2. Foam Stability (*FS*)

*FS* can be described as the period of time in which the foam maintains its original properties when formed [1]. *FS* is mainly influenced by temperature and pH, by the nature of the proteins and their interaction, which should produce a viscous, elastic and air-impermeable film around each air bubble to stabilize the system [1]. *FS* values were visually monitored over a period of time of 90 min. *FS* values for FG, PPI and MIX are presented in Figure 4. We can observe that *FS* did not improve for all the samples after US treatment. Both FG and PPI sonicated samples showed a slight decrease in *FS* after 90 min, range from 13.33 to 12.0% and from 106.0 to 100.67%, respectively. MIX reached the same value around 114% for both untreated and sonicated samples. Furthermore, MIX foams present higher values, over the whole period considered, compared to FG and PPI foams. This may suggest a possible protein-protein interaction on the film formation around the bubbles. Globally, sonicated samples present similar values to the controls, without any statistical difference. As can be seen from the graphs, the treatment provided higher *FS* values during the firsts 25 min, while they dropped below controls values in the remaining 60 min. In particular, FG curves decreased steadily, approaching *FS* of 0% over the considered period, while both PPI and MIX stabilized above 100% during 90 min, presenting a similar behavior. 

Overall, the results are in accordance with the study conducted by Xiong et al. [10]. where foaming properties of PPI were evaluated after different conditions of ultrasound treatment. In this study, *FS* values of sonicated PPI were significantly higher than those of untreated PPI, but only during the first 10 min of evaluation, whereas after 20 min, the authors found no significant difference between the two samples. This trend can be attributed to the partial unfolding of proteins induced by US, which allows them to rapidly adsorb at the A-W interface of the foam, leading to a greater stability within the first 10 min. However, in a larger period of time, protein molecules are desorbed from the interface due to the increase of air bubble sizes, a condition that leads them to interact with other desorbed molecules forming new aggregates. The aggregation phenomenon is even more pronounced within sonicated proteins because of the higher degree of exposed hydrophobic groups. This mechanism drastically reduces protein stability effect at the surface, reducing the overall *FS* values after a period of 20 min [10,24]. 

#### 3.5.3. Liquid Fraction (*LF*)

*LF* represents the water fraction trapped in the Plateau border between air bubbles [1]. During the time, gravitational forces cause the drainage of water from the borders resulting in a gradual collapse of the foam. Monitoring *LF* changes over time therefore can assess protein capacity of trapping water and an overall evaluation of foam stability. *LF* values were visually monitored over a period of time of 90 min for each sample and are hereby presented as the average of three replicates. *LF* values for FG, PPI and MIX are presented in Figure 5. We can observe that US treatment decreased *LF* percentages of FG and PPI samples after 90 min, going from 8.0 to 5.33% and from 8.67 to 2.0%, respectively. On the other hand, sonicated MIX sample showed a value of 9.33%, greater than 2% compared to the untreated one. Generally, all the samples tent to 0% as expected, decreasing rapidly in the firsts 20 min and steadily for the rest of the time. The obtained results are in general accordance with the study conducted by Morales et al. [23] where US were employed on soy protein isolate in an attempt of modifying their foaming functionalities. Even for soy proteins, it was not observed a significant change of liquid drainage from a foam after US treatment. However, sonicated samples showed a slight increase in the drainage velocity, and thus a faster decrease of *LF* %, as occurred in FG and PPI samples of this study. 

#### 3.5.4. Turbiscan Tower

Immediately after the whipping process, FG, PPI and MIX foams were transferred into a 55 mm high tube and loaded into the Turbiscan Tower instrument. In particular, the two-phase separation occurs, where the drained liquid drops to the bottom whereas the foam stays at the top. Samples were run during 90 min, the same period of time considered by the visual analysis of their foaming properties. Over the time, 60 scans were performed (1 every 1.5 min) measuring the backscattering value (BS) all over the height of the tube. BS values over time, as obtained from the instrument, are presented in Appendix A. 

BS% is caused by air bubbles of the foam. BS% value decrease over time due to foam aging and its gradual collapse. Foam collapse is mainly described by two effects: liquid drainage and bubbles coalescence. Liquid drainage effect is represented by the continuous shift of BS peak over the height of the tube sample, while coalescence is described by the steady decrease of BS% value. Both of these effects are influenced by the size of the air bubbles. Over time, bubbles tend to coalesce together and their size increase until it reaches the size limit. Over this limit the bubble bursts and the liquid that was trapped is now released, generating a drainage phase at the bottom and decreasing BS%. Turbiscan software, based on the Mie equation, allows the automatic computation of mean diameter of air bubbles from the BS level on the foam phase. The results are presented in Figure 6. 

All of the samples start with a similar mean diameter, generally over 115 µm, and during the period of time considered they evolve in a different way. In particular, FG both untreated and sonicated increase rapidly in bubble size until they burst around 30 min with a diameter above 500 µm. On the other hand, PPI and MIX samples present a similar trend: they steadily increase their mean diameter, overcoming FG limit size and reaching a size around 600 µm, without bursting over the 90 min considered. Furthermore, we can observe that the US treatment changed the evolution of the bubbles size of FG and PPI samples, by increasing their diameter. In particular, it caused the collapse of FG sonicated sample earlier than the untreated one. Nevertheless, this change did not cause the earlier burst for sonicated PPI, which maintained a good stability over time. Regarding MIX samples, the US treatment did not change significantly their behaviour and showed the same evolution over time. The results are overall in agreement with the visual analysis on foaming properties. Especially with *FS* from which it results that FG samples stability constantly decreases with foam aging, here shown with the bubbles burst, and the US treatment even decreased it. While PPI and MIX samples reached a good stability of the foam and the US treatment did not improve it significantly. 

### 3.6. Confocal Laser Scanning Microscopy (CLSM) 

CLSM was employed to evaluate the distribution and behaviour of proteins at the A-W interface and to better understand foam microstructure. However, due to the relative instability of FG foams, only PPI and MIX samples were analyzed by this technique. Indeed, FG foams drained too much liquid into the sample well, covering all of the air bubbles and hiding them from the microscope vision.

From the movies recorded by the instrument, frames of the sample bubbles were collected and Figure 7, was designed. From Figure 7, it can be clearly seen the fluorescent film layer made from protein aggregates, as suggested by DLS measurements. Ultimately, the protein layer was confirmed with the image comparison with a bathroom soap foam, free of proteins in its composition. PPI and MIX foam frames were then compared and no significant difference was seen between untreated and sonicated proteins. Moreover, MIX protein layers did not differ from PPI, suggesting that pea proteins only contributed to the film formation when mixed with fish proteins, giving a similar result to PPI foam. Therefore, it was not possible to detect an interaction between FG and PPI aggregates at the A-W interface. 

On the contrary, Jarpa-Parra et al. [25] employed CLSM on foams stabilized by legumin-like protein and polysaccharides and the technique succeed to visualize an interaction between the two biopolymers due to different fluorescent excitation wavelengths. The authors concluded that the interaction was allowed by the opposite charges of the polymers and the formation of weak electrostatic attractions between them. 

### 3.7. Interfacial Tension (IFT) 

IFT at the A-W interface was measured with the pendant drop method. The instrument was calibrated monitoring the IFT of water before measure. IFT values as a function of time of FG, PPI and MIX solutions is presented in Figure 8. After 1000 s we can observe a decrease only for FG samples after US treatment. Probably this could be caused by the different protein size of FG and PPI. In addition, FG stabilized at 52.2 ± 0.3 mN/m whereas PPI and MIX present stable values at 48.9 ± 0.1 mN/m and 47.2 ± 0.2 mN/m. Therefore, combining the two proteins allowed to reach the lowest IFT values. A similar study was realized by Xiong et al. [10]. The authors applied 30 min of US treatment at different amplitudes (30, 60, 90%) and observed a general decrease of IFT values for sonicated PPI when compared to control. These results could be explained by the improved affinity of PPI to the dispersed phase, as more hydrophobic regions were exposed after the sonication. Indeed, an increase in hydrophobicity can result in a reduction of the energy barrier at the A-W interface, allowing a facilitated adsorption [26]. In our study, after 5 min of treatment, no significant differences were observed for both PPI and MIX samples when compared to their controls. Perhaps the treatment did not last enough to increase the exposed hydrophobic regions of PPI aggregates and thus to change the rate adsorption at the interface. Moreover, as explained by ζ-potential values, the US treatment increased surface charges of both PPI and MIX samples resulting in higher electrostatic barriers at the A-W interface and thus causing the decrease of their adsorption rate [27]. 

## 4. Conclusions and Perspectives 

The interaction between FG and PPI in solution and at A-W interface, before and after US treatment was investigated. We focused on bulk rheology and the interfacial tension contribution of the protein solutions and then we evaluated the respective foaming properties. The treatment was responsible for a viscosity decrease of both FG and PPI samples, whereas MIX showed the same flow properties, in between FG and PPI. IFT value at the A-W interface of the solutions revealed that US treatment only affected FG proteins while PPI and MIX did not show a significant change. However, lower IFT values were obtained when the proteins were combined. Regarding the foaming properties, four main parameters were evaluated: foam capacity (*FC*), foam stability (*FS*), liquid fraction (*LF*) and bubble size evolution over time. Results showed that this specific ultrasound treatment (275 W, 5 min) only improved *FC* of FG while PPI and MIX were not significantly affected. *FS* was not improved by the sonication for FG and PPI, while MIX samples had the highest *FS* values during all the tests conducted, suggesting a possible interaction and improvement in stabilizing the foam system compared to the single protein foams. In addition, US allowed MIX sample to have higher *LF* values than its control meaning it could slow the liquid drainage and the foam to dry over the time, while FG and PPI on the other hand slightly decreased in value with the treatment. In the end, bubble size evolution over time confirmed the better stability achieved by both PPI and MIX while FG bubbles rapidly grew in size until they busted, causing the foams to collapse. Moreover, US treatment not only did not improve stability of FG samples but caused an earlier collapse of their foams. Based on these considerations we can conclude that the specific US treatment employed cannot significantly improve these protein sources foaming properties. Overall, MIX samples presented the most encouraging interfacial properties, improving slightly the characteristics of PPI only, as if PPI and FG had developed a weak interaction between them. To better inspect a possible interaction, a visual analysis of the protein layer around each air bubble in foam was employed using CLSM, but unfortunately, no differences were noticed between PPI and MIX layers. Further investigations are therefore needed to demonstrate this possible interaction between FG and PPI. In addition, different US treatments could be applied to the samples as presented by Xiong et al., 2018 [10], changing the combination of power and time. In fact, the treatment submitted in this study seemed to be too weak to significantly affect PPI and MIX behavior at the A-W interface. Finally, CLSM could also be employed to calculate the thickness of the protein layer and the aggregates adsorption ratio at the A-W interface. 

## Figures and Tables

**Figure 1 foods-11-00659-f001:**
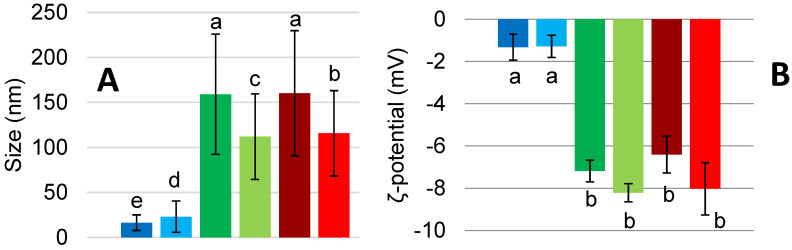
(**A**) Hydrodynamic diameter (nm) and (**B**) ζ-potential (mV) of FG (■ untreated, ■ treated), PPI (■ untreated, ■ treated) and MIX (■ untreated, ■ treated). Values are here reported as the average of three replicates and their standard deviation is shown in the error bars. a–e, different letters show significant differences (*p* < 0.05).

**Figure 2 foods-11-00659-f002:**
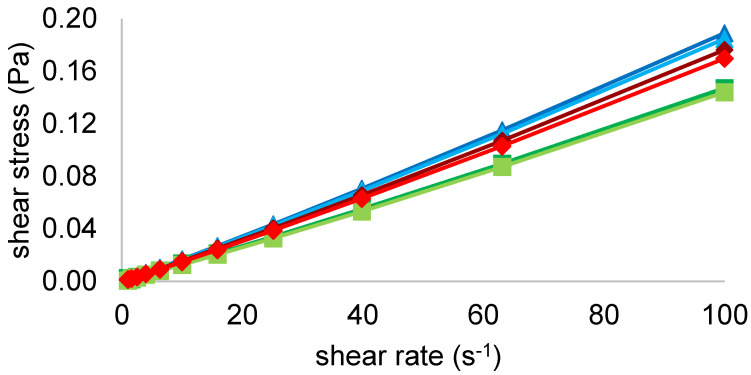
Shear stress (Pa) versus shear rate (s^-1^) of FG (■ untreated, ■ treated), PPI (■ untreated, ■ treated) and MIX (■ untreated, ■ treated) at 20 °C.

**Figure 3 foods-11-00659-f003:**
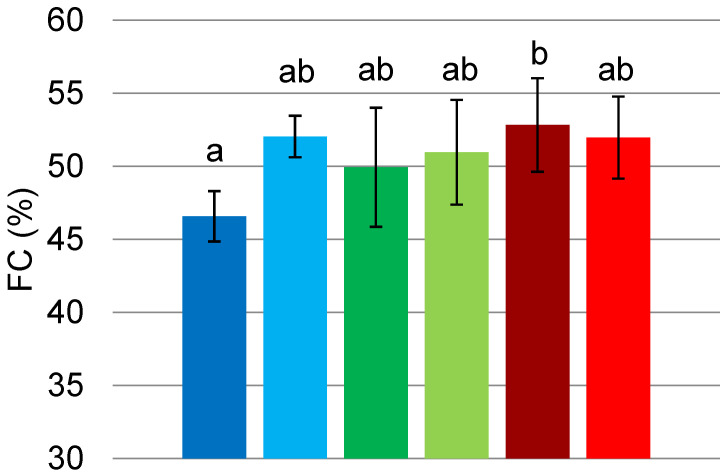
*FC* (%) of FG (■ untreated, ■ treated), PPI (■ untreated, ■ treated) and MIX (■ untreated, ■ treated). Values are here reported as the average of three replicates and their standard deviation is shown in the error bars. Different letters show significant differences (*p* < 0.05).

**Figure 4 foods-11-00659-f004:**
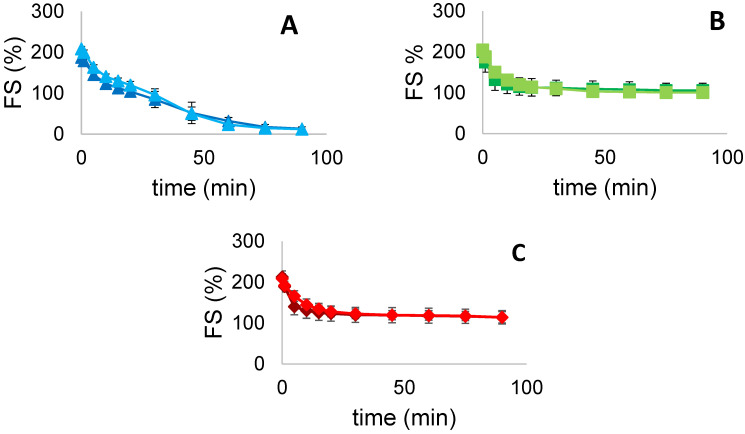
*FS* (%) over time (min) for (**A**) FG (■ untreated, ■ treated), (**B**) PPI (■ untreated, ■ treated) and (**C**) MIX (■ untreated, ■ treated). Values are here reported as the average of three replicates and their standard deviation is shown in the error bars.

**Figure 5 foods-11-00659-f005:**
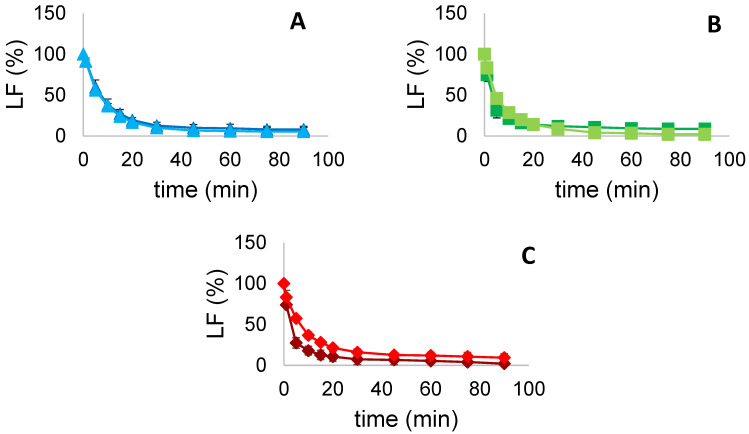
*LF* (%) over time (min) for (**A**) FG (■ untreated, ■ treated), (**B**) PPI (■ untreated, ■ treated) and (**C**) MIX (■ untreated, ■ treated). Values are here reported as the average of three replicates and their standard deviation is shown in the error bars.

**Figure 6 foods-11-00659-f006:**
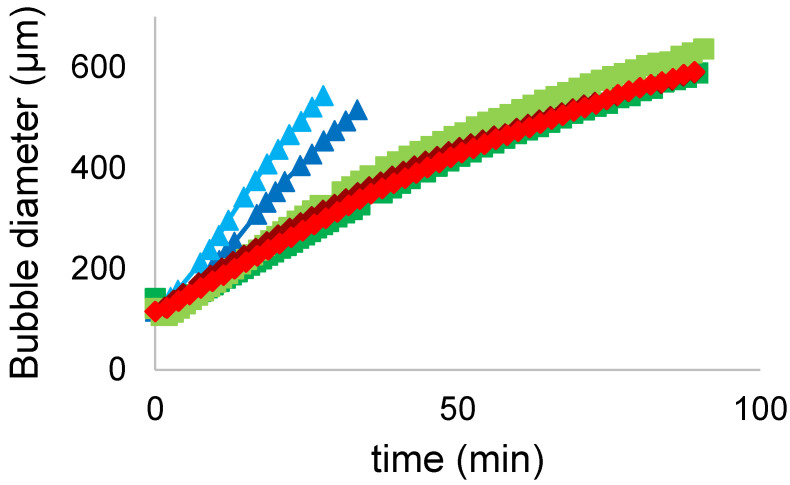
Bubble mean diameter (µm) evolution over time (min) of FG (■ untreated, ■ treated), PPI (■ untreated, ■ treated) and MIX (■ untreated, ■ treated).

**Figure 7 foods-11-00659-f007:**
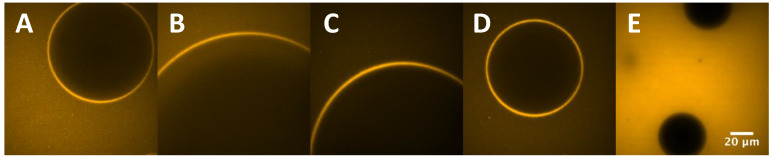
CLSM images of PPI (**A**,**B**) and MIX (**C**,**D**) samples. To discriminate the fluorescent protein layer around each air bubble, a comparison with soap bubbles with no protein content (**E**) has been carried out.

**Figure 8 foods-11-00659-f008:**
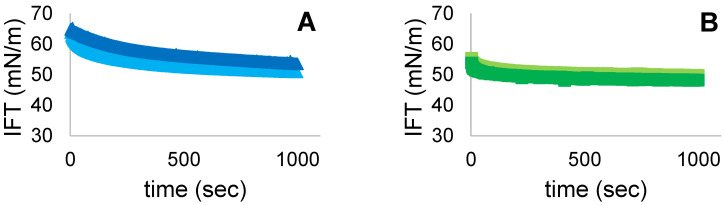
IFT values (mN/m) over time (sec) for (**A**) FG (■ untreated, ■ treated), (**B**) PPI (■ untreated, ■ treated) and (**C**) MIX (■ untreated, ■ treated).

**Table 1 foods-11-00659-t001:** pH values of the samples (Fish gelatin FG, pea protein isolate PPI, and their mixture MIX) as the average of three repeated measurements before and after a high intensity ultrasound treatment (275 W, 5 min).

pH	FG	PPI	MIX
Untreated	7.13 ± 0.1 ^a^	7.34 ± 0.2 ^b^	7.27 ± 0.1 ^d^
Sonicated	7.14 ± 0.1 ^a^	7.24 ± 0.1 ^c^	7.15 ± 0.3 ^e^

Means with different superscripts in each column indicate significant differences (*p* < 0.05).

**Table 2 foods-11-00659-t002:** Molecular weight distribution of FG, PPI and MIX before and after sonication.

Sample		Fraction 1	Fraction 2	Fraction 3	Fraction 4
		Mw (kDa)			
FG	Untreated	33.2			
	Sonicated	29.8			
PPI	Untreated	879.4	164.4	61.2	44.1
	Sonicated	820.3	167.6	67.1	49.4
MIX	Untreated	1048.1	208.9	102.0	84.9
	Sonicated	2269.5	428.4	188.7	136.7

**Table 3 foods-11-00659-t003:** Power–law model parameters of all the samples at 20 °C.

Power Law	*m* (mPa.s) *	*n* **	R^2^
FG untreated	1.3	1.1	0.99
FG sonicated	1	1.1	0.95
PPI untreated	1.5	1	0.98
PPI sonicated	1	1.1	0.98
MIX untreated	1.2	1.1	0.99
MIX sonicated	1.3	1	0.99

* Consistency coefficient; ** Flow behavior index.

## Data Availability

Data is contained within the article or Appendix A.

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
