# Peer review of "Interaction between Fish Skin Gelatin and Pea Protein at Air-Water Interface after Ultrasound Treatment"

_foods, 2022, doi:10.3390/foods11050659_

Round 1

Reviewer 1 Report

Please, improve the font of all Figures

Author Response

Reviewer 1

Please, improve the font of all Figures.

We understand the Reviewer and the quality of the figure is improved. In any case, before publishing the raw data (graphs and pictures) will be sent to publishing service.

Reviewer 2 Report

This work investigated the interaction between fish skin gelatin (FG) and pea protein isolate (PPI) at the air-water interface before and after ultrasound treatment (US). However, there are obvious shortcomings in the experimental design, the discussion is poor and cannot support the conclusion. In addition, the ultrasonic treatment is not an obvious improvement.

  1. What is the basis for the selection of ultrasonic conditions?
  2. How to distinguish the role of two proteins in the water-air interface?
  3. What is the specific interaction between the two proteins?
  4. The surface hydrophobicity of the particles is critical for adsorption at the interface, why not test it?
  5. Interfacial tension (IFT): Whether 1000s can explain the adsorption of protein at the interface?
  6. Table 1, Table 2 Data needs to be analyzed for significance.
  7. Line 289 All the changes are minor, does that explain the results?
  8. Lines 302-304 Similar studies exist, what is the innovation of this paper?
  9. Lines 466-469 This part can be placed in the method.
  10. Line 500 Is there a rationale for the authors to think that ultrasound is not sufficient to expose hydrophobic groups? Surface hydrophobicity data are required.

Author Response

Reviewer 2

This work investigated the interaction between fish skin gelatin (FG) and pea protein isolate (PPI) at the air-water interface before and after ultrasound treatment (US). However, there are obvious shortcomings in the experimental design, the discussion is poor and cannot support the conclusion. In addition, the ultrasonic treatment is not an obvious improvement.

1. What is the basis for the selection of ultrasonic conditions?

The selection of those specific ultrasonic conditions is based on the previous study (non-published and conducted by the same group). 

2. How to distinguish the role of two proteins in the water-air interface?

To distinguish the role of the two proteins adopted at the air-water interface, we analyzed the interfacial properties of the single protein suspensions, which represented the controls. The results were then compared with the results obtained for a ratio (50-50%) of these two proteins.

3. What is the specific interaction between the two proteins?

We assumed that the two proteins presented a synergistic behavior at the air-water interface. In specific, PPI helped retaining FG at the interface developing weak hydrophobic interaction between them. We assumed this based in particular on the results of Sec-Malls, where protein fractions Mw was increased when mixed. However, we agree with the reviewer that other approaches will be necessary to better understand the contribution at air-water interface, such interfacial rheology and/or ellipsometry.

4. The surface hydrophobicity of the particles is critical for adsorption at the interface, why not test it? We agree with the reviewer however, the surface hydrophobicity of FG and PPI proteins (after and before ultrasound treatment), have been already evaluated by the same Group and in specific by Mar Vall-llosera (DOI: 10.1007/s11483-020-09655-7), co-author in this study. The study found out that PPI presents higher hydrophobicity values of a factor of 10 than FG, and a low frequency coupled with high intensity ultrasound treatment, as the one used in this study, further increased their values.

5. Interfacial tension (IFT): Whether 1000s can explain the adsorption of protein at the interface?

We understand the point of the reviewer. This specific time was considered to be enough to understand the adsorption of proteins at the interface. In specific, FG and PPI exhibited a rapid decrease of their values during the first seconds of acquisition followed by a stabilization after 500s. Data were also collected for further time over 1000s showing good stability and reproducibility, thus it was chosen to present only the values until 1000s.

6. Table 1, Table 2 Data needs to be analyzed for significance.

We agree with the reviewer and we analyze the data in Table 1. However, Table 2 regards the molecular weight distribution of FG and PPI. We analyzed the samples twice and no difference were observed.   

7. Line 289 All the changes are minor, does that explain the results?

We understand the point of the reviewer. In the case of FG and PPI, we observe a decrease in Mw. When we Mix these two proteins, an increase in Mw is revealed. This suggests that weak interactions between FG and PPI fractions have occurred when the proteins were mixed and that they have further increased after the sonication. 

8. Lines 302-304. Similar studies exist, what is the innovation of this paper?

The innovation of this paper regards the development of an innovative colloidal system stabilized by the mixture of two sustainable hydrocolloids: FG and PPI. In specific, FG represents a food industry by-product which could be employed in food formulations. Moreover, to improve its functional properties, FG has been combined with an emerging protein: PPI. Regarding the methodology, Turbiscan Tower, as well as CLSM have been employed to characterize the foaming properties of this complex. In the literature, we can find only few examples of their employment in a food system. Lastly, this innovative mixture of proteins could expand the offer of environmental-friendly food products in the future.

9. Lines 466-469. This part can be placed in the method.

That specific part has been moved into the methodology of CLSM, in the new lines 211-214.

10. Line 500. Is there a rationale for the authors to think that ultrasound is not sufficient to expose hydrophobic groups? Surface hydrophobicity data are required.

We understand the reviewer. As we explain previously (point 4), hydrophobicity was tested by the same group, in a previous study. We observed an increasing in hydrophobicity after ultrasound treatment. In this Manuscript, and in specific in the paragraph 3.7, we think that our ultrasound treatment hasn’t been enough to increase the hydrophobicity of the proteins in order to see some effect at air-water interface.

Reviewer 3 Report

Overall:

Minor English editing MUST be made during revisions.

The authors present a study conducted to investigate the interaction between fish skin gelatin and pea protein isolate at the air-water interface before and after ultrasound treatment. Overall this work is interesting and relevant to foods.

The authors should make minor edits (English editing, clarification of figures/tables, and statistical methods) prior.

Abstract:

“finally” in line 24 can be omitted

Change “PPI aggregates” in line 27 to “PPI aggregate”

Introduction:

Line 36 remove “between each other’s”

Materials and methods:

Statistical analysis: were any post hoc tests conducted to determine which samples were different? The specific test used should be clarified.

Table 1 could be improved by adding details (e.g., n), clarifying in a footnote the abbreviations and the ultrasound conditions, etc. although these are mentioned elsewhere in the manuscript.

Figure 1A: are the error bars correct? There appears to be large overlap between samples with statistical groupings of A, B and C. Clarify in the figure or caption the n and what the error bars represent (e.g., standard deviation, standard error, 90% CI, etc.) Note, similar additions should be made for other applicable figures (e.g., Fig 3) as well.

Figure 7- The caption should be corrected here. There is an additional note (line 485) that should be adjusted.

Author Response

Reviewer 3

Minor English editing MUST be made during revisions.

The authors present a study conducted to investigate the interaction between fish skin gelatin and pea protein isolate at the air-water interface before and after ultrasound treatment. Overall this work is interesting and relevant to foods.

The authors should make minor edits (English editing, clarification of figures/tables, and statistical methods) prior.

Abstract:

“finally” in line 24 can be omitted.

Finally has been removed.

Change “PPI aggregates” in line 27 to “PPI aggregate”.

The word has been changed.

Introduction:

Line 36 remove “between each other’s”.

It has been removed.

Materials and methods:

Statistical analysis: were any post hoc tests conducted to determine which samples were different? The specific test used should be clarified.

The statistical methodology has been clarified.

Table 1 could be improved by adding details (e.g., n), clarifying in a footnote the abbreviations and the ultrasound conditions, etc. although these are mentioned elsewhere in the manuscript.

Table 1 has been improved and clarified.

Figure 1A: are the error bars correct? There appears to be large overlap between samples with statistical groupings of A, B and C. Clarify in the figure or caption the n and what the error bars represent (e.g., standard deviation, standard error, 90% CI, etc.) Note, similar additions should be made for other applicable figures (e.g., Fig 3) as well.

The error bars represented in figure 1A are correct; the reason for the large standard deviation of the sample is commented already in the text. The figures have been clarified.

Figure 7- The caption should be corrected here. There is an additional note (line 485) that should be adjusted.

The caption of that figure has been corrected.

Round 2

Reviewer 2 Report

Accept.